# Production of Hydrophobic Microparticles at Safe-To-Inject Sizes for Intravascular Administration

**DOI:** 10.3390/pharmaceutics17010064

**Published:** 2025-01-06

**Authors:** Francisca L. Gomes, Francisco Conceição, Liliana Moreira Teixeira, Jeroen Leijten, Pascal Jonkheijm

**Affiliations:** 1Laboratory of Biointerface Chemistry, Department of Molecules and Materials, Faculty of Science and Technology, Technical Medical Centre and MESA+ Institute, University of Twente, 7522NB Enschede, The Netherlands; f.l.fernandesgomes@utwente.nl; 2Leijten Laboratory, Department of BioEngineering Technologies, Faculty of Science and Technology, Technical Medical Centre, University of Twente, 7522NB Enschede, The Netherlands; 3Department of BioEngineering Technologies, Advanced Organ Bioengineering and Therapeutics, Faculty of Science and Technology, Technical Medical Centre, University of Twente, 7522NB Enschede, The Netherlands; f.conceicao@utwente.nl (F.C.); l.s.moreirateixeira@utwente.nl (L.M.T.); 4Organ-on-Chip Centre Twente, MESA+ Institute, University of Twente, 7522NB Enschede, The Netherlands

**Keywords:** microparticles, polycaprolactone, emulsification, injectability, blood capillary

## Abstract

**Background/Objectives:** Hydrophobic microparticles are one of the most versatile structures in drug delivery and tissue engineering. These constructs offer a protective environment for hydrophobic or water-sensitive compounds (e.g., drugs, peroxides), providing an optimal solution for numerous biomedical purposes, such as drug delivery or oxygen therapeutics. The intravascular administration of hydrophobic microparticles requires a safe-to-flow particle profile, which typically corresponds to a maximum size of 5 µm—the generally accepted diameter for the thinnest blood vessels in humans. However, the production of hydrophobic microparticles below this size range remains largely unexplored. In this work, we investigate the fabrication of hydrophobic microparticles at safe-to-inject and safe-to-flow sizes (<5 µm) for intravascular administration. **Methods:** Polycaprolactone microparticles (PCL MPs) are produced using a double-emulsification method with tip ultrasonication, for which various production parameters (PCL molecular weight, PCL concentration, type of stabilizer, and filtration) are optimized to obtain particles at sizes below 5 µm. **Results:** We achieve a PCL MP size distribution of 99.8% below this size limit, and prove that these particles can flow without obstruction through a microfluidic model emulating a thin human blood capillary (4.1 µm × 3.0 µm width × heigh). **Conclusions:** Overall, we demonstrate that hydrophobic microparticles can be fabricated at safe-to-flow sizes using a simple and scalable setup, paving the way towards their applicability as new intravascular injectables.

## 1. Introduction

Hydrophobic microparticles are extensively used in pharmaceutics and tissue engineering as drug carriers or scaffold components. Numerous techniques have been used in the fabrication of hydrophobic microparticles, such as batch emulsification [1,2,3,4,5], microfluidics droplet generation [6,7,8,9], membrane emulsification [10,11], spray drying [12], electrospraying [13,14], or supercritical emulsion extraction [15]. Batch emulsification is the simplest and most frequently employed strategy in hydrophobic microparticle production, consisting of the emulsification of an organic phase, rich in the hydrophobic material, and an aqueous phase, rich in stabilizers, with resort to an agitation source (e.g., magnetic stirring, ultrahomogenization, or ultrasonication). Emulsification can be performed as a single emulsion, also named oil-in-water (*o*/*w*), or double emulsion, known as water-in-oil-in-water (*w*/*o*/*w*). Both result in polydisperse droplets that originate from hydrophobic microparticles after solvent elimination, with the difference being the absence or presence of an inner aqueous core, respectively. Single-emulsion particles are often used to encapsulate lipophilic molecules, whereas double-emulsion particles are often used with hydrophilic molecules. This simple, inexpensive, and versatile technique can be tuned for particle size through the agitation source and tuned for cargo encapsulation through the single-/double-emulsion modality.

Numerous advances in biomedicine have significantly benefitted from the use of hydrophobic microparticles, often as drug delivery systems or engineered tissue components. These particles have shown outstanding results in oral and nasal drug administration [16,17,18], bone regeneration [19,20], cartilage regeneration [21], dermal fillings [10,22], and self-oxygenating scaffolds [5,23] The most commonly employed materials are polycaprolactone (PCL), polylactic acid (PLA), poly(lactic-co-glycolic acid) (PLGA), or copolymers thereof, with nature-based alternatives including zein [24,25] or keratin [26,27].

To safely inject particles through an intravascular route, particles should be able to flow undisturbed through the blood vasculature of the patient. Human blood capillaries exhibit typical diameters of 8 µm [28] and minimum diameters of 3.5–4.5 µm [29,30,31], and therefore the generally accepted maximum particle size for injection is 5 µm. However, producing hydrophobic microparticles at 1–5 µm remains a challenge, primarily due to the requirement of high-precision equipment to produce uniform droplets at this size range and finely tuned interfacial tension to prevent droplet coalescence and particle aggregation.

In this work, we investigated the possibility of fabricating hydrophobic microparticles at sizes below 5 µm for safe intravascular administration. Polycaprolactone (PCL) was used to produce spherical microparticles through a conventional double-emulsification method with solvent evaporation, using tip ultrasonication. To obtain a particle size below 5 µm, different production parameters (PCL molecular weight, PCL concentration, and type of stabilizer) were screened and analyzed in terms of <5 µm size compliance. Particles with the highest percentage of sizes below 5 µm were then flown through a microfluid-based model simulating a thin human blood capillary to determine their flowability profile.

## 2. Experimental Section

### 2.1. Materials

Polycaprolactone (PCL) of different average molecular weights (M_n_ 80 kDa, M_n_ 45 kDa, and M_n_ ~10 kDa) was purchased from Sigma-Aldrich (Zwijndrecht, The Netherlands). Polyvinyl alcohol of a molecular weight (MW) of 205 kDa (PVA, Mowiol^®^ 40-88, M_w_ 205 kDa), gelatin (porcine skin, type A, bloom 300), HEPES (≥99.0%), dichloromethane (DCM, ACS reagent, ≥99.5%), and methanol (HPLC grade, ≥99.9%), were purchased from Sigma-Aldrich. Ethanol (absolute, ≥ 99.5%) was purchased from VWR Chemicals (Leuven, Belgium). Ultrapure water was obtained from Millipore (Leuven, Belgium).

### 2.2. Double-Emulsification Method

For the double-emulsification method, 0.6 mL of ultrapure water was first emulsified in 6 mL of a PCL solution in dichloromethane, made at different polymer concentrations (3%, 5%, and 10% *w*/*v*) and different polymer molecular weight (80 kDa, 45 kDa, 10 kDa). The emulsification was performed on ice using a Fisherbrand™ Q500 Sonicator (Fisher Scientific, Ochten, The Netherlands) at an amplitude of 20% for 1 min, with a pulse of 2 s on, 2 s off. Then, 3.3 mL of this emulsion were added to 50 mL of either a PVA solution (0.3% *w*/*v*) or a gelatin solution (1% *w*/*v*), and the second emulsion was then produced again under tip sonication, at an amplitude of 25% for 4 min with a pulse of 2 s on 2 s off. The double emulsion was stirred at 600 rpm overnight to evaporate the dichloromethane, and the resulting suspension was washed once in methanol, once in ethanol, and twice in ultrapure water at ~15,000× *g* for 25 min. Particle filtration was conducted under vacuum using mixed cellulose ester membranes of 5 µm cutoff (Merck-Millipore, Leuven, Belgium).

### 2.3. Dynamic Light Scattering and Zeta Potential

Dynamic light scattering (DLS) and zeta potential (ZP) measurements were conducted on a ZetaSizer Nano ZS (Malvern, Almelo, The Netherlands), at a scattering angle of 173°, at 25 °C. PCL MPs were redispersed in 1 mL of HEPES buffer (10 mM) at concentrations below 0.5 mg mL^−1^ and measured five-fold.

### 2.4. Scanning Electron Microscopy

Scanning electron microscopy (SEM) measurements of PCL MPs were performed on a JSM 7610 F-Plus microscope (JEOL, Nieuw-Vennep, The Netherlands) at an acceleration voltage of 1.5 kV and a working distance of 8 mm. Particle suspensions were cast on a small silicon substrate and allowed to air-dry for analysis.

### 2.5. Particle Size Analysis

Particle suspensions were prepared in water at comparable dilution factors and analyzed using an Olympus BH2 optical microscope (Olympus, Hoofddorp, The Netherlands). Each condition was imaged at different regions of interest (ROIs) and a fixed number of ROIs were used to compare the particle count between samples. Microscope images were analyzed using Fiji software (https://fiji.sc/, accessed on 12 December 2024) [32] by using the thresholding and particle analysis functions. The data were further processed using Origin software (2023 SRO), with the use of the Statistics functions to determine the coefficient of variation (CV) and the frequency counts of particles below 5 µm.

### 2.6. Flowability Assay

Particle flowability was assessed on a microfluidics model of a thin human blood capillary (diameter 3.5–4.5 µm) [29,30]. Chip design was performed using CleWin software (version 5.4.30.0, WieWeb). Master molds were produced based on micropatterned SU-8. Briefly, h100i orientation silicon wafers (Okmetic, Vantaa, Finland) were spin-coated with an SU-8 5 negative photoresist (Micro Resist Technology GmbH, Berlin, Germany), producing an average layer height of the central chamber of ~5 μm. The SU-8 photoresist was then patterned by exposure to UV light with a 365 nm longpass filter using an EVG 620 mask aligner (EV Group, St. Florian, Austria). The patterned wafers were then developed in RER600 (Fuji Film, Tilburg, The Netherlands) followed by spraying, spinning, and drying. Finally, patterned wafers were washed with isopropanol and dried under a nitrogen stream.

Microfluidics chips were produced through PDMS-based soft lithography using a Sylgard^®^ 184 silicone elastomer kit (VWR Chemicals, Leuven, Belgium). PDMS and the curing agent were mixed at a weight ratio of 5:1, cast on the photomask mold, and cured for 2 h at 60 °C. Each PDMS chip was then prepared and cleaned prior to plasma bonding. Channel width was determined from the average width of 6–8 sections of each channel, as observed through an EVOS FL microscope (Thermo Fisher, Ochten, The Netherlands). Channel height was determined from scanning electron micrographs of a section of the channel, obtained with a Jeol JSM-IT 100 microscope (JEOL, Nieuw-Vennep, The Netherlands). For the flowability assay, chips were first hydrated with 5 µL and 1 µL of water on the inlet and outlet, respectively, until full-channel hydration. A suspension of PCL MPs at low concentrations was then stained with 3 µg mL^−1^ DiOC_6_, pipetted into the chip inlet (5 µL) along with a small droplet of water (1 µL) on the outlet, and imaged over one hour after injection and three days after injection on an EVOS FL microscope.

## 3. Results

### 3.1. Production of PCL MPs Through Double Emulsification

The double-emulsification process was initiated by the addition of an aqueous phase to a PCL solution in dichloromethane, followed by their emulsification through tip ultrasonication. This first emulsion was then emulsified in a second aqueous phase, rich in stabilizers, using the same ultrasonication technique. The second emulsion was allowed to stir overnight to facilitate dichloromethane evaporation, during which a restructuring of the polymer chains occurred and the final microparticles were formed. The complete process is schematized in Figure 1.

### 3.2. Effect of PCL MW, Type of Stabilizer, and Filtration

To investigate whether the double emulsification method could be used to produce PCL MPs at sizes below 5 µm, we explored the effects of polymer molecular weight (M_w_), type of stabilizer, and filtration on particle size. To this end, we used PCL M_w_s of 80 kDa, 45 kDa, or 10 kDa, based on the three most commonly reported values for PCL MP fabrication [2,15,33,34], and PVA or gelatin as stabilizers, based on their non-toxic and biocompatible profile [35,36]. It is important to note that a PVA with a low hydrolysis rate (87-89%) was used, as the stabilization of organic droplets in water often requires a water-soluble stabilizer with affinity towards the organic phase. Stabilizer concentrations (0.3% *w*/*v* PVA, 1% *w*/*v* gelatin) were further determined based on previous reports on PCL MP production [5,23]. We observed that a lower PCL M_w_ resulted in a lower particle size, with similarly low size distributions for PCL 45 kDa and 10 kDa (Figure 2, dark bars). For these M_W_s, no significant differences were observed between the particle size distributions of 0.3% PVA or 1% gelatin (Figure 2, dark bars). Nevertheless, all batches presented a fraction of particles larger than 5 µm, and therefore a membrane filtration step was further included. However, upon filtration, we observed significant particle loss in all batches, with little effect in the removal of the fraction of large particles (Figure 2, light bars).

Given the similar size distributions obtained for both MWs and both types of stabilizers, we further quantified the percentage of particles below 5 µm in these conditions. Both M_w_s (45 kDa and 10 kDa) produced a remarkable percentage of particles within this size limit (>98%), with the combinations of 45 kDa–PVA and 10 kDa–Gelatin reporting the highest percentages, at 99.3% and 99.1%, respectively (Figure 3, dark dots). After filtration, the resulting size distributions were only marginally reduced for PCL 45 kDa and 10 kDa, but were considerably lowered for PCL 80 kDa (Figure 3, light dots). This effect is likely a result of vacuum-induced aggregation, which is more prominent in smaller hydrophobic particles (PCL 45 kDa, 10 kDa) than larger particles (PCL 80 kDa). Nevertheless, after filtration, a percentage of 100% was achieved in the batch 45 kDa–Gelatin (Figure 3, green circle), ultimately demonstrating that PCL MPs could be produced through batch emulsification at sizes entirely below 5 µm. Collectively, these results show that PCL MWs of 45 kDa and 10 kDa are preferred for small particle production, regardless of the type of stabilizer, with size distributions at 99.3% and 99.1% below 5 µm, respectively. Moreover, filtration proved essential in achieving a size distribution entirely below 5 µm, although the considerable particle loss associated with this step warrants alternative solutions.

### 3.3. Effect of PCL Concentration

To determine the effect of PCL concentration on particle size, we further compared the size distribution of particles made with 3%, 5%, and 10% *w*/*v* PCL using dynamic light scattering and zeta potential. We hypothesized that a lower PCL concentration (3% and 5%) could result in a further reduction in particle size, as compared to the currently used concentration (10%), and ultimately avoid the need for particle filtration. With this in mind, PCL MPs were produced at the three different concentrations using PCL 10 kDa and 0.3% *w*/*v* PVA. Here, PVA was preferred over gelatin due to the fact that PCL MPs produced at M_w_ 10 kDa in 1% gelatin were considerably more prone to aggregation than those produced using 0.3% PVA (Appendix A).

From all tested concentrations, PCL 3% resulted in the lowest average particle size, lowest polydispersity index (PdI), and highest zeta potential modulus (Table 1, Figure 4a,b). The Z-average sizes of PCL 3% were determined at a range of 1.0–3.5 µm, demonstrating a marked reduction as compared to PCL 10% (3.5–7.5 µm, Figure 4a,b). PCL 3% also presented the lowest value for PdI and zeta potential, indicating the highest particle stability in suspension. We further quantified the percentage of particles below 5 µm for PCL 3% using optical microscopy (Figure 4c). The particle size distribution was calculated at 99.8% below 5 µm, based on three replicate batches, with an average size of (1.4 ± 0.9) µm and a coefficient of variation of 66.2% (Figure 4e,f). Moreover, scanning electron microscopy confirmed a majority of particles at sizes below 5 µm (Figure 4d). Overall, these results represent the highest reduction in microparticle size, demonstrating that 3% PCL, 10 kDa, and 0.3% PVA are the most promising parameters for PCL MP production at safe-to-inject sizes using batch emulsification.

### 3.4. Flowability Tests

To confirm whether the optimized small PCL MPs could flow safely through human blood vessels, we observed the behavior of particle flow in a thin capillary-like microchannel (Figure 5a). The microchannel displayed an average width of 4.1 ± 0.3 µm based on optical microscopy imaging, whereas channel height was calculated at 3.0 µm based on SEM imaging (Figure 5b). This platform represents a minimal yet reliable model for the narrowest blood vessels in humans, having been recently employed in particle flowability tests due to their facile production, freedom from ethical constraints, and higher level of similarity to human capillaries [30,37].

PCL MPs were injected in the thin microcapillary at low concentrations, demonstrating a safe flow with no obstructions to the channel (Figure 5c). An undisturbed PCL MP flow was also observed in a wider model of a human blood capillary, representing the typical size of these blood vessels (~8 µm width, Appendix A). However, upon complete flow of the particle suspensions, we often observed particle retention in the microcapillaries (Figure 5d), an effect derived from the hydrophobic interactions between PCL MPs and the PDMS walls of the channel. Importantly, this effect was not linked to particle size, since particles were visibly smaller than the width of the channels, and therefore it is likely that particle retention would be lower in a more hydrophilic, biomimetic channel. Moreover, particle flowability could be further improved by endowing PCL MPs with an anti-fouling hydrophilic coating (e.g., a synthetic lipid membrane), which would render them more bioinert in both capillary-on-chip and in vivo models.

Collectively, we demonstrated that PCL MP suspensions, produced at 3% PCL 10 kDa and 0.3% PVA, were sufficiently safe to flow in a human blood capillary model at low concentrations, providing a core material for intravascularly injectable products. Future studies should focus on the application of an anti-fouling coating on the hydrophobic microparticles and a subsequent reinvestigation of their behavior under flow.

## 4. Discussion

### 4.1. Effect of Production Parameters on Particle Size

The concentration and M_w_ of PCL were here shown to play a significant role on the size distribution of the PCL microparticles. Lower PCL concentrations yielded a lower particle diameter, mainly due to the lower viscosity of the solutions [38], which facilitated the emulsification process and resulted in smaller droplets. A lower PCL M_w_ also resulted in a lower particle diameter, which was observed for other hydrophobic polymers (PLGA [39,40]) and most likely related to the more condensed rearrangement of shorter polymeric chains in the droplet. However, due to the similarity in the observed size distributions for microparticles made of 45 kDa and 10 kDa PCL, it appears that M_w_ only plays a significant role on PCL MP sizes above 45 kDa. Of note, these results were obtained with tip ultrasonication, a high-intensity agitation source, which might not produce the same correlation between M_w_ and particle size as a low-intensity agitation source (e.g., magnetic stirring).

Regarding the type of stabilizer, both 0.3% PVA and 1% gelatin yielded a similar particle size distribution. However, PCL MPs produced at a MW of 10 kDa were considerably more prone to aggregation in 1% gelatin as opposed to 0.3% PVA (Appendix A), suggesting that the latter is a more suitable stabilizer for low-MW polymers. This phenomenon strengthens the preference for PVA, the most commonly reported stabilizer in the production of PCL MPs [2,38].

Membrane filtration also proved important in achieving a fully safe-to-inject particle batch. However, the inclusion of this step led to a significant particle loss, which would pose a major drawback in potential production upscaling. The following optimization process led to a particle size distribution of 99.8% below 5 µm, successfully reducing the need for particle filtration. In fact, according to regulatory guidelines, the optimized particle formulation could be deemed safe to inject without filtration, provided that a specific particle concentration was not exceeded. This aspect is further elaborated in the next subsection.

### 4.2. Applicability of PCL MPs as Intravascular Injectable Particles

According to the current safety practice in parenteral drug administration, products up to 100 mL are considered safe to administer if the microparticle content of the container is lower than 6000 (particles ≥ 10 µm) [41,42]. For products above 100 mL, the product is considered safe to administer if the microparticle content is lower than 25 particles mL^−1^, (particles ≥ 10 µm) [42]. Considering a product above 100 mL, such as most intravenous solutions, a PCL MP size distribution of 99.8% below 5 µm (i.e., 0.2% above 5 µm) would correspond to a maximum particle concentration of 1.25 × 10^4^ microparticles mL^−1^ to be considered safe to inject intravascularly (Figure 6a,b). Considering the current microparticle concentration, we calculated the mass of a single PCL MP as 1.64 × 10^−3^ pg, and plotted the estimated cargo load of the total PCL MP suspension at different encapsulation efficiencies (Figure 6c). The range of pg mL^−1^ falls short of multiple drug delivery systems in the market, which round the cargo load of mg mL^−1^ (e.g., Doxil^®^) [43] or mg per dose (e.g., Eligard^®^, Risperidal^®^ Consta) [44]. Nevertheless, this cargo load can be beneficial in other applications, such as oxygen therapeutics, where certain oxygen-carrying cargo (e.g., nano-peroxides) provide their most effective action at low doses.

### 4.3. Challenges in Manufacturing Hydrophobic Microparticles as Intravascular Injectables

In this study, hydrophobic microparticles were produced via double emulsification in a simple, inexpensive, and highly scalable manner, at sizes predominantly below 5 µm. While this constitutes an advance in regard to other production protocols [6,45], there are considerable disadvantages in using double emulsification in large-scale microparticle manufacturing.

Good manufacturing practices (GMPs) require product uniformity, specifically particle monodispersity and inter-batch reproducibility. These aspects are exceptionally difficult to achieve through batch emulsification, and more so in the case of hydrophobic microparticles at sizes below 5 µm. Firstly, batch emulsification often results in highly polydisperse microparticles due to limited control over the rate of solvent evaporation. Secondly, the inter-batch variability of double emulsification is highly reliant on several conditions, such as vat dimensions, ultrasonication time, tip depth, or evaporation time. Thirdly, the maximum size limit of 5 µm, required for safe intravascular administration, means it is difficult to achieve monodispersity using hydrophobic materials. An evident solution to this problem would be the use of nanoparticles, which fit the size requirement and are typically monodisperse; however, their functionality for some applications, such as erythrocyte engineering, would be affected by their reduced cargo load. Finally, batch emulsification requires a post-filtration step to achieve fully safe injectability (100% particle size < 5 µm), but this step often encompasses major particle loss, consequently reducing the applicability of PCL MPs in bioengineering. Omitting the filtration step would be an option for the current formulation; however, the maximum accepted particle concentration would be considerably reduced (1.25 × 10^4^ particles mL^−1^), thereby leading to the same limitation in applicability.

To sum up, double emulsification provides a simple small-scale solution to the fabrication of hydrophobic microparticles, but not on a large industrial scale. Under these conditions, particles should be fabricated monodispersely, in ultrahigh throughput, and at sizes entirely below 5 µm, to produce a viable and universal safe-to-inject product. With this in mind, alternative fabrication methods should be sought.

### 4.4. Potential Solutions

Microfluidic droplet generation has gained traction in recent years due to its potential to produce small (<50 µm) monodisperse microparticles. The advantage of being a continuous, automated method enables better control over intra-batch uniformity and inter-batch reproducibility, thus facilitating GMP compliance. However, this technique has the disadvantages of a limited throughput and a challenging production of ultrasmall microparticles (<5 µm) in a monodisperse manner. Considerable effort has been dedicated to increasing the throughput of microfluidics (e.g., through chip parallelization [6,45] or in-air microfluidics [46]), but attempts at producing ultrasmall microparticles are still scarce. Recently, the commercialization of monodisperse PLGA microparticles in a high-throughput manner at a size of 2 µm was announced, offering a promising pathway for the production of small hydrophobic microparticles [47].

Template-based microparticle fabrication is another possible method to produce hydrophobic engineered erythrocytes. Template bases are compatible with different materials, can be extended and reused for ultrahigh throughput, and offer an inherently high particle monodispersity. One of the most promising template techniques, PRINT^®^ (particle replication in nonwetting templates), is based on a flexible microparticle template with a well-defined size and shape, loaded with a polymer precursor to yield uniform microparticles [48]. A high throughput can be achieved using roll-to-roll technology, where the template is recirculated and recycled for continuous production [48]. Moreover, it has the advantage of being GMP-compliant [48], having been used to produce 6 µm sized particles [49].

Membrane emulsification also constitutes a high-throughput approach to the production of small microparticles. In short, two immiscible phases are placed in a reservoir, separated by a uniform porous membrane, which acts as a droplet template as one phase is pressurized onto the other. This method is compatible with hydrophobic polymers and has been used in the production of monodisperse microparticles below 5 µm [50,51,52].

## 5. Conclusions

PCL MPs were successfully produced through double emulsification at a size distribution of 99.8% below 5 µm. For different production conditions, the addition of a filtration step at 5 µm cut-off improved the particle size distribution to 100% below 5 µm. Particles at sizes of 99.8% below 5 µm demonstrated an undisturbed flow through a thin blood capillary, strongly indicating a safe-to-flow profile through the human blood vasculature. While representing a significant step forward in the production of intravascularly injectable microparticles, the acceptable particle concentrations for safe administration were severely limited using the current production method. Moreover, GMP requirements for microparticle size, monodispersity, and high throughput would be more accurately met with the use of alternative fabrication methods. Future studies should explore microfluidics droplet generation, template-based methods (PRINT^®^), or membrane emulsification to produce sub-5 µm microparticles according to these requirements. Subsequent functional tests would provide a final assessment of their potential use as intravascular injectables in drug delivery, oxygen therapeutics, or erythrocyte engineering. Nevertheless, we here demonstrate the possibility of producing hydrophobic microparticles at safe-to-inject sizes using simple laboratory equipment, consequently paving the way for new proof-of-concept studies in nanomedicine.

## Figures and Tables

**Figure 1 pharmaceutics-17-00064-f001:**
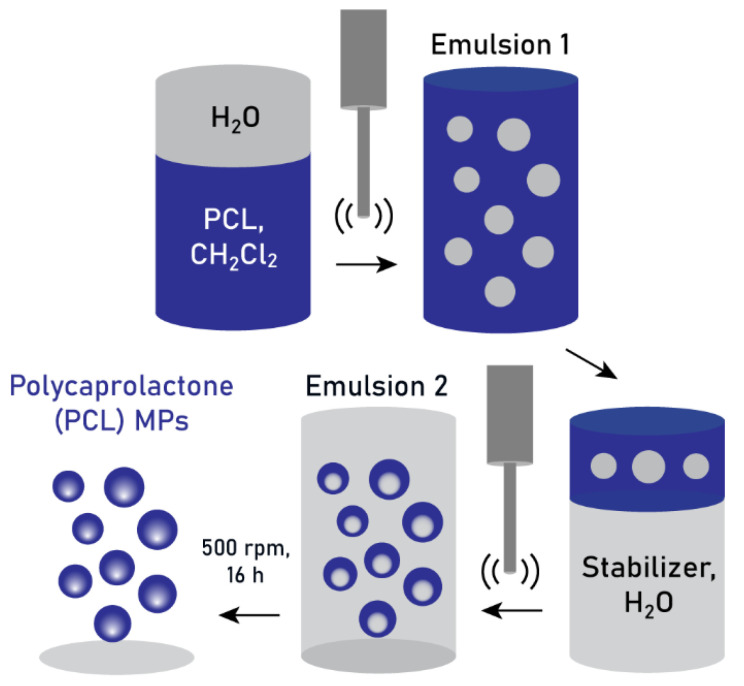
Schematic depiction of the double-emulsification process, with tip ultrasonication, to produce PCL MPs.

**Figure 2 pharmaceutics-17-00064-f002:**
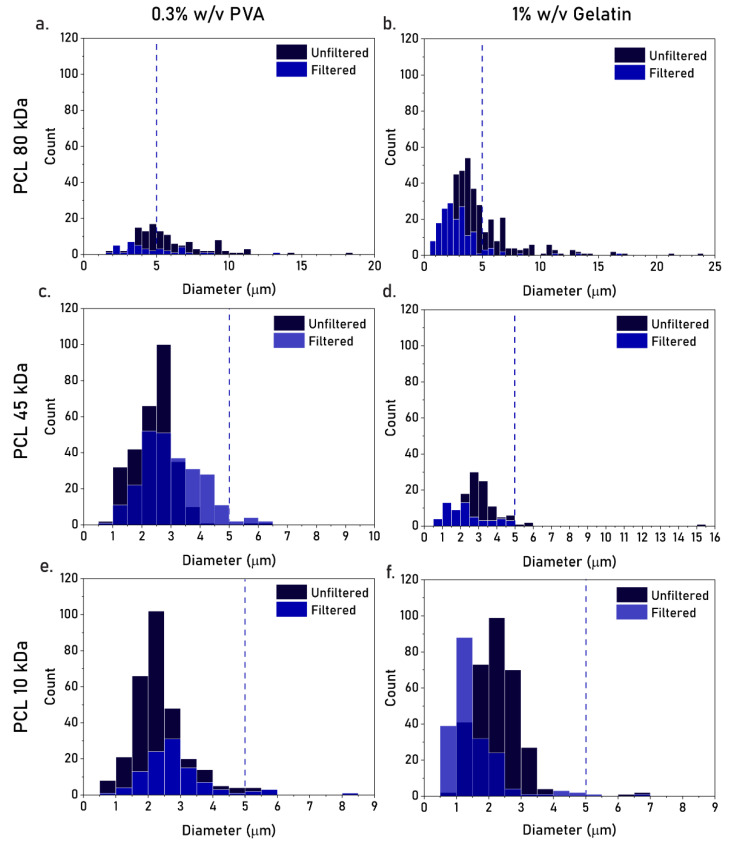
(**a**–**f**) Screening for the effect of PCL MW, type of stabilizer, and membrane filtering on the size distribution of PCL MPs. The conditions were achieved with PCL 80 kDa (**a**,**b**), 45 kDa (**c**,**d**) or 10 kDa (**e**,**f**), and stabilized with 0.3% *w*/*v* PVA (**a**,**c**,**e**) or 1% *w*/*v* gelatin (**b**,**d**,**f**). The dark bars indicate no filtering, the light bars indicate 5 µm cut-off filtering. Dashed lines indicate the 5 µm cut-off.

**Figure 3 pharmaceutics-17-00064-f003:**
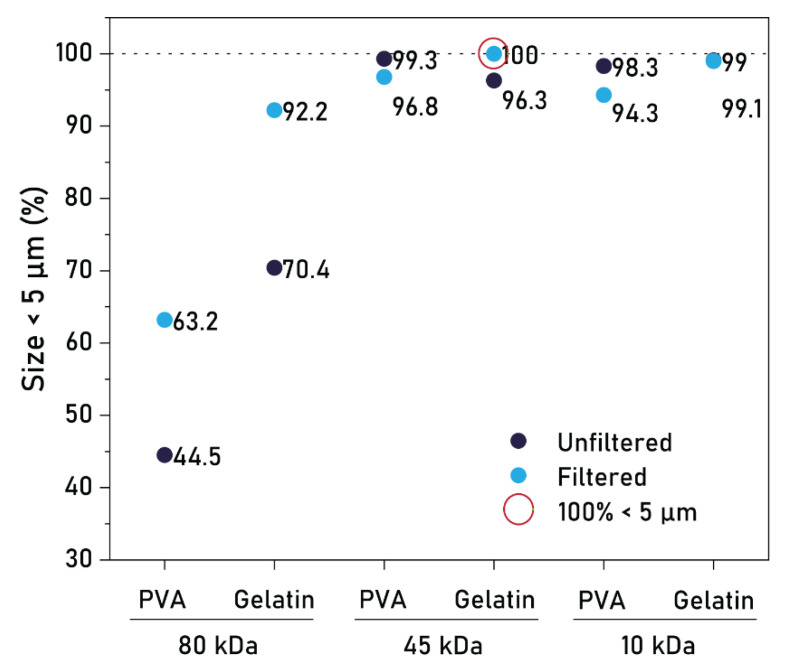
Percentage of PCL MPs below 5 µm in diameter for different production conditions. Dark dots indicate unfiltered batches, light dots indicate filtered batches. Green circle indicates a size distribution of 100% below 5 µm.

**Figure 4 pharmaceutics-17-00064-f004:**
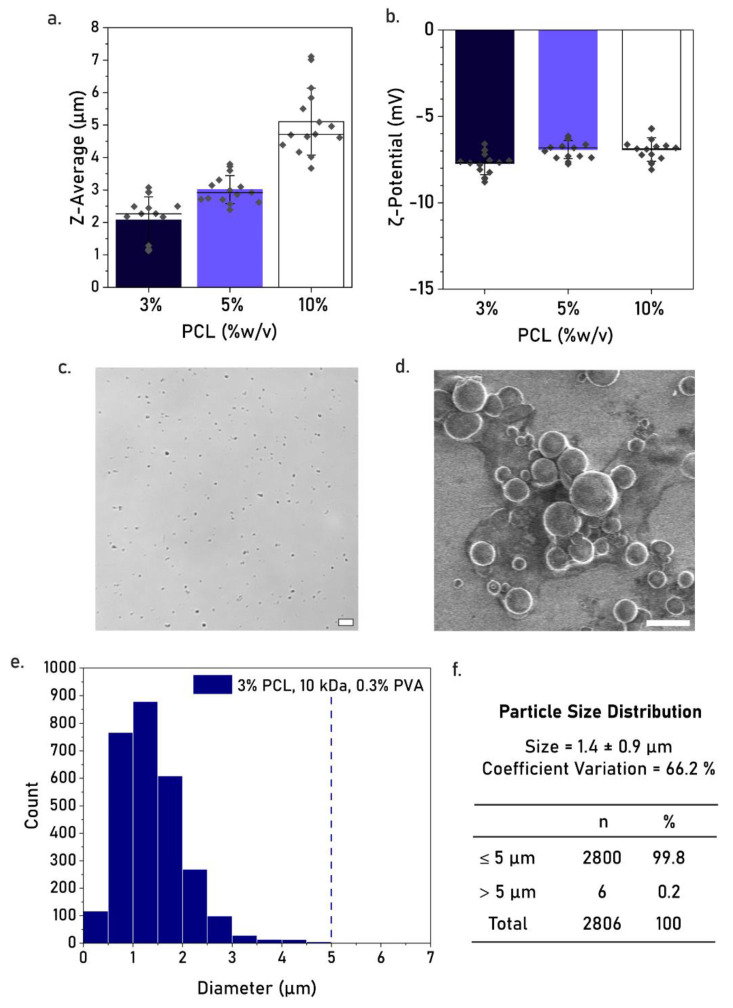
(**a**,**b**) Screening for the effect of PCL concentration (3%, 5%, 10% *w*/*v*) on size distribution (**a**) and zeta potential (**b**) of PCL MPs produced using 0.3% *w*/*v* PVA and PCL 10 kDa. (**c**) Optical micrograph of PCL MPs. Scale bar equals 10 µm. (**d**) The scanning electron micrograph of PCL MPs. Scale bar equals 5 µm, the original SEM image can be found in Appendix A. (**e**) The particle size distribution of PCL MPs based on optical microscopy. Data collected for n = 3 batches. (**f**) The summary of particle size, coefficient of variation, and percentage of particles below 5 µm. The dashed line indicates 5 µm cut-off.

**Figure 5 pharmaceutics-17-00064-f005:**
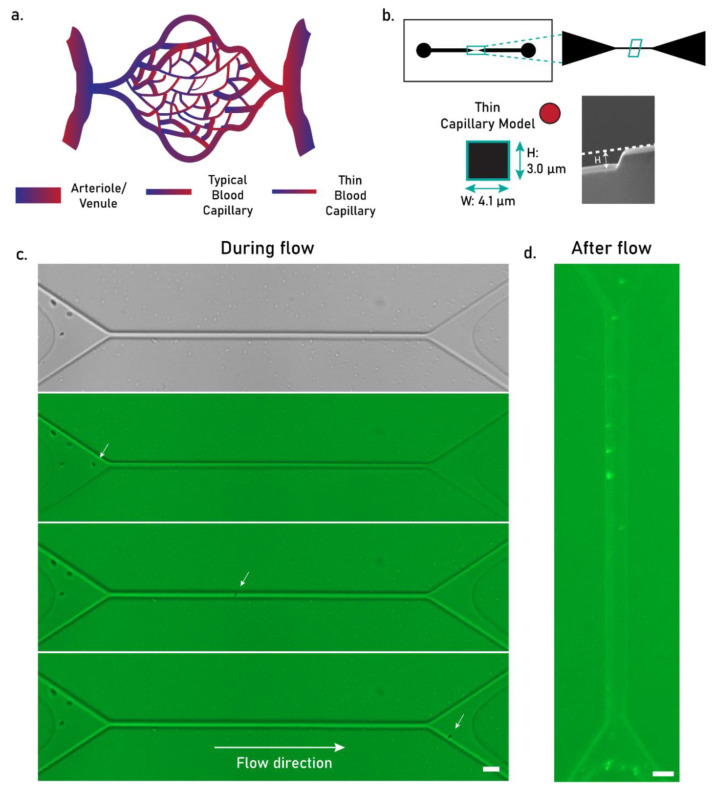
(**a**) Schematic depiction of a blood capillary network. (**b**) Schematic depiction of chip design for a microfluidics model of a thin blood capillary. (**c**) Epifluorescence image of a PCL MP (stained with DiOC_6,_ in green) during flow. The arrows indicate particles. The scale bar equals 10 µm. (**d**) An epifluorescence image of a microfluidics model simulating a thin blood capillary upon complete flow of PCL MPs (stained in green). The scale bar equals 10 µm.

**Figure 6 pharmaceutics-17-00064-f006:**
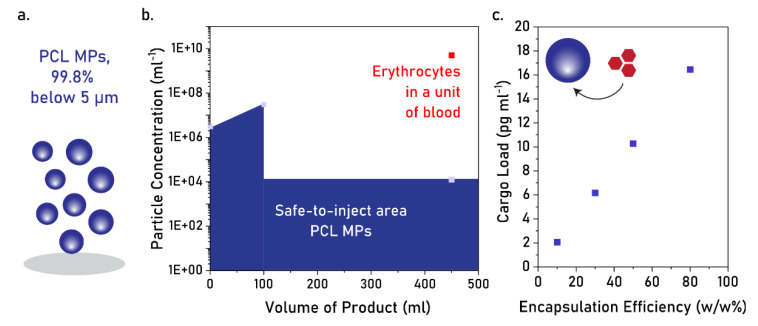
(**a**) Schematic depiction of optimized PCL MPs. (**b**) Safe-to-inject concentrations for PCL MPs at different product volumes. Concentration of erythrocytes in a unit of blood represented as reference. (**c**) Calculated cargo load for different encapsulation efficiency rates.

**Table 1 pharmaceutics-17-00064-t001:** Dynamic light scattering and zeta potential results for the triple-batch comparison of PCL MPs produced at different polymer concentrations. Data represented as mean ± standard deviation for n = 3 batches, each averaged for n = 5 measurements.

PCL Concentration (*w*/*v*)	PCL MW	Stabilizer	Z-Average Size (nm)	PdI	Zeta Potential (mV)
3%	10 kDa	0.3% *w*/*v* PVA	2275 ± 388.3	0.589 ± 0.062	−7.73 ± 0.36
5%	10 kDa	0.3% *w*/*v* PVA	3010 ± 145.9	0.867 ± 0.057	−6.92 ± 0.40
10%	10 kDa	0.3% *w*/*v* PVA	5104 ± 403.4	0.857 ± 0.038	−6.92 ± 0.54

## Data Availability

The original contributions presented in this study are included in the article/Appendix A. Further inquiries can be directed to the corresponding author(s).

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
