# Peer review of "Production of Hydrophobic Microparticles at Safe-To-Inject Sizes for Intravascular Administration"

_pharmaceutics, 2025, doi:10.3390/pharmaceutics17010064_

Round 1

Reviewer 1 Report

Comments and Suggestions for Authors

1.As shown in the Figure 5. a, human capillaries have a reticular multi-level structure. However, the microfluidics chips used in this paper has a single-level structure. Therefore, I think that experiments on connecting chips in series should be added. In this way, the real physiological phenomena of human capillaries can be better simulated.                     

2.From the results of table 1, we can see that the size distribution range is very wide. The PDI values are all greater than 0.5 and even reach 0.85. This indicates that the particle distribution range is very large and uneven. Therefore, in addition to controlling the particle size to be no more than 5 microns, should this paper add particle size range control? Because particles of uneven sizes are more likely to block capillaries.

Author Response

Dear Reviewer,

Thank you for reviewing our manuscript. Please find our point-by-point reply below.

1.As shown in the Figure 5. a, human capillaries have a reticular multi-level structure. However, the microfluidics chips used in this paper has a single-level structure. Therefore, I think that experiments on connecting chips in series should be added. In this way, the real physiological phenomena of human capillaries can be better simulated.   

Thank you for the remark. Indeed, it would have been more physiologically accurate to test particle flow on a multi-level capillary-like structure. At the beginning of the study, we extensively researched the following network-like capillary options:

  1. https://micronit.com/eor-physical-rock-vertical-crack.html
  2. https://microfluidics.creative-biolabs.com/vessel-on-a-chip-model-development-service.htm
  3. An in-house developed 3D capillary network, inspired by: https://onlinelibrary.wiley.com/doi/10.1002/adma.202308949 
  4. An in-house developed capillary network based on cotton candy, inspired by: https://pmc.ncbi.nlm.nih.gov/articles/PMC11333037/ 
  5. Customized microfluidic chip by design. 

However, we have found that these platforms could not accurately mimic the thinnest blood vessel in humans (3.5-4.5 µm in diameter), and customized chips by design is beyond the aim and reach of this manuscript. We therefore opted for a single-channel structure for this study, which we believe gives sufficient insights in the potentialities of our achievements.   

2. From the results of table 1, we can see that the size distribution range is very wide. The PDI values are all greater than 0.5 and even reach 0.85. This indicates that the particle distribution range is very large and uneven. Therefore, in addition to controlling the particle size to be no more than 5 microns, should this paper add particle size range control? Because particles of uneven sizes are more likely to block capillaries.

Thank you for the comment. In section 4.3 (p. 10, l. 315-316) we explained that one of the issues of hydrophobic microparticle production is their highly polydisperse size distribution. This is especially visible when using the current production techniques. We aimed for the highest number of analyzed particles throughout the study, reaching a representation as high as n = 2806 for the final round of particle optimization (Fig.4f). In the result section, we also included the coefficient of variation (66.2%), a parameter of particle size range. We further discuss the improvement of particle size control throughout section 4.4, with the description of alternative production methods with a lower chance of size polydispersity.

Best regards,

The authors

Reviewer 2 Report

Comments and Suggestions for Authors

This is a concise study dealing with the production of hydrophobic microparticles of P.S. less than 5 um so that it would be apt for intravascular injections.

I recommend its publication after performing the following:

1- Did the authors perform any physical stability studies for the prepared hydrophobic microparticles? This is v. imp for intravascular injections as the particles may increase in size or aggregate blocking the blood vessels.

2- Also, do the authors performed or have any evidence for the response of the prepared microparticles to sterilization ?

Author Response

Dear Reviewer,

Thank you for reviewing our manuscript. Below please find a point-by-point reply to each comment.

  1. Did the authors perform any physical stability studies for the prepared hydrophobic microparticles? This is v. imp for intravascular injections as the particles may increase in size or aggregate blocking the blood vessels.

Thank you for the remark. Indeed, physical stability is essential for intravascular injection, in particular the prevention of particle aggregation. We have indeed observed that such small hydrophobic microparticles tend to aggregate in water, especially after dispersion in isopropyl alcohol and redispersion in aqueous buffer. To address this issue, we have developed a surface treatment/coating using lipids to improve colloidal stability. The lipid coating of particles (how to do that) is the focus of our manuscript that we will submit elsewhere. The current results are valid to show that we can synthesize injectable blood capillary sized polymer particles and flow them through a capillary line.

  1. Also, have the authors performed or have any evidence for the response of the prepared microparticles to sterilization ?

As expected, we can disperse the microparticles in isopropyl alcohol/ethanol and in gradual dilutions of isopropyl alcohol/ethanol with aqueous buffer.

Best regards,

The authors

Round 2

Reviewer 1 Report

Comments and Suggestions for Authors

Please check the whole text. There are some errors in typesetting, citation or spelling in this article. For example, page 5 line 174, there is a citation error.